# Purification and molecular characterization of phospholipase, antigen 5 and hyaluronidases from the venom of the Asian hornet (*Vespa velutina*)

**Rafael I. Monsalve**[1]*, **Ruth Gutiérrez**[1], **Ilka Hoof**[2], **Manuel Lombardero**[1]

**1** CMC Research and Development, ALK-Abelló S.A., Madrid, Spain, **2** Global Research, ALK-Abelló A/S, Hørsholm, Denmark

* Rafael.monsalve@alk.net

**Data Availability Statement:** The new Sequence data are presented in the manuscript (all the complete sequences extracted from the transcriptomic databases, as well as their experimental confirmation after the purifications of

## Abstract

The aim of this study was to purify potential allergenic components of *Vespa velutina* venom, the yellow legged Asian Hornet, and perform a preliminary characterization of the purified proteins. Starting from the whole venom of *V. velutina*, several chromatographic steps allowed to purify the phospholipase (named Vesp v 1), as well as the antigen 5 (Vesp v 5, the only allergenic component described as such so far). The two hyaluronidase isoforms found (Vesp v 2A and Vesp v 2B) cannot be separated from each other, but they are partially purified and characterized. Purity of the isolated proteins in shown by SDSPAGE, as well as by the results of the N-terminal sequencing. This characterization and nLC-MS/MS data provide most of the sequence for Vesp v 1 and Vesp v 5 (72 and 84% coverage, respectively), confirming that the whole sequences of the isolated natural components match with the data available in public transcriptomic databases. It is of particular interest that Vesp v 1 is a glycosylated phospholipase, a fact that had only described so far for the corresponding allergen components of *Dolichovespula maculata* and *Solenopsis invicta*. The availability of the complete sequences of *Vespa velutina* components permits comparison with homologous sequences from other Hymenoptera. These data demonstrate the higher similarity among the species of the genera *Vespa* and *Vespula*, in comparison to *Polistes* species, as it is especially observed with the hyaluronidases isoforms: the isoform Vesp v 2A only exists in the former genera, and not in *Polistes*; in addition, the most abundant isoform (Vesp v 2B) exhibits 93% sequence identity with the Ves v 2 isoform of *Vespula vulgaris*. Finally, the isolated components might be useful for improving the diagnosis of patients that could be allergic to stings of this invasive Asian hornet, as it has been the case of an improved diagnosis and treatment of other Hymenoptera-sensitized patients.

## Introduction

The "yellow-legged"-Asian hornet (*Vespa velutina nigrithorax*) is an invasive species that was originally introduced in the Iberian Peninsula through France, having been identified in Spain

the proteins). The protocols followed in this manuscript are explained in the Materials and Methods section, as well as with the corresponding citations of the manuscripts that first performed the corresponding studies for, e.g. purification, enzymatic activity measurement, etc. The data from our experimental determinations have been deposited in public repositories and are freely accessible. The LTQ Orbitrap Velos raw files and files with data of all the peptides identified for the purified Vesp v 5 and Vesp v 1 are freely accessible and are indexed as ProteomeXchange dataset PXD015381 and included in the MassIVE server. Moreover, the complete sequences for Vesp v 5, Vesp v 1, Vesp v 2A and Vesp v 2B data can be accessed from the UniProt Knowledgebase with the following accession numbers: P0DMB9 for Vesp v 5, C0HLL3 for Vesp v 1, C0HLL4 for hyaluronidase A (Vesp v 2A) and C0HLL5 for hyaluronidase B (Vesp v 2B).

**Funding:** All the authors who are sending this paper, as mentioned in our affiliations, are employees of "ALK-Abelló", belonging to the Research and Development (CMC R&D) in Madrid-Spain, and Research department (Global Research) in Horsholm-Denmark. The "Funder" ALK-Abelló provided support in the form of salaries for authors RIM, RG, IH, ML, but did not have any additional role in the study design, data collection and analysis, decision to publish, or preparation of the manuscript. The specific roles of these authors are articulated in the 'author contributions' section.

**Competing interests:** The potential competing interest of the authors, in their research and preparation of the article for publication can be considered the following: - We have no "non-financial" competing interests, that may affect our presented research - all the authors are employees of ALK-Abelló (as mentioned we all belong to Research and Development Departments of this company); this is the only funding source for our work. None of them have stocks or shares. - The "funder" ALK-Abelló has had no intervention in the study design; collection, analysis, and interpretation of data; writing of the paper; and/or decision to submit for publication. Besides, our potential competing interest do not alter our adherence to PLOS ONE policies on sharing data and materials.

for the first time in 2010 [1]. Since then, it has extended its presence through the north of Spain, up to the north of Portugal, causing socio-economic concerns due to its important expansive capacity, also considering that they frequently attack honey-bee hives to feed their nests. Solutions for this invasion are being considered in different ways in several European countries [2–5]. From the allergologic point of view, it is also a concern, as demonstrated in the North of Spain [6], when the first cases of anaphylactic shocks were described, and when "antigen 5", from the venom of these Asian hornets, was identified as the most important allergen in a group of eight sensitized patients.

The data we have obtained from these purified components, confirm the whole sequence data available in transcriptomic databases since 2015 [7], and this allows a thorough comparison with homologous components from other Hymenoptera. Moreover, the purified components will be used to demonstrate their relevance as allergens, by Component Resolved Diagnosis (CRD), in studies performed in platforms that can measure specific IgE in the sera of allergic patients ([8,9]), which are useful to differentiate which can be the sensitizing species for each patient, as we also published in the past ([10]), differentiating on *Vespula* versus. *Polistes* sensititzation.

The nomenclature used for naming the purified components, will consider what is suggested for allergenic proteins by the WHO/IUIS (World Health Organization / International Union of Immunological Societies) Committee ([11,12]), considering that only the antigen 5 (named Vesp v 5) has been described as such so far; the phospholipase (Vesp v 1) and hyaluronidases (Vesp v 2) must still be considered as potential allergenic components.

In summary, in this work we isolate the main potential allergenic components of the *V.velutina* venom, and show the potential interest on the availability of these purified components: their characterization allow a closer understanding of similarities among different Hymenoptera venom components, and will surely be useful in a better diagnosis and treatment of allergic patients to stings of this Asian hornet.

## Materials and methods

The relevant natural venom components (A1 phospholipase, antigen 5 and hyaluronidases) were purified from lyophilized *Vespa velutina* venom sac extract of individual hornets collected in Europe (ALK Source Materials Inc., Spring Mills, U.S.A.; batch 01071301AH). The purification was performed as previously described [13,14], and in a similar manner as performed with other natural allergens from *Vespula vulgaris* and *Polistes dominula* [10]. The purified proteins were analysed by SDS–PAGE (Novex-Tricine, 10–20% acrylamide, Invitrogen Life Technologies, Carlsbad, CA, U.S.A.) and silver- or Coomassie blue-stained, depending on the needs of the analysis (SDSPAGE was silver stained, according to [15]). The enzymatic activity of phospholipase and hyaluronidases was verified when needed by the methods of Habermann [16] and Richman and Baer [17], respectively, using *Apis mellifera* venom (ALK Source Materials Inc., Spring Mills, U.S.A.) preparation as reference.

The purity and identity of the purified proteins was also confirmed, according to [18], by N-terminal sequencing analyses (direct analysis of the protein in solution), performed at CIB Protein Chemistry Service (CSIC, Madrid, Spain).

In the case of Vesp v 1 and Vesp v 5 additional nLC-MS/MS (nano Liquid Chromatography tandem mass spectrometry) analyses were performed from the bands extracted from an SDSPAGE stained with Coomassie blue (Colloidal Blue Staining, LC6025, Invitrogen Life Technologies, Carlsbad, CA, U.S.A.): this protein identification by nLC-MS/MS was carried out in the Proteomics and Genomics Facility (CIB-CSIC, Madrid—Spain), a member of ProteoRed-ISCIII network, according to the method described in [19]. For the MS analysis,

Peptides were trapped onto a Acclaim PepMap 100 (Thermo Fisher Scientific Inc., Waltham, MA U.S.A.) precolumn, and then eluted onto a column Acclaim PepMap 100 C18 column, inner diameter 75 μm, 25 cm long, 3 μm particle size (Thermo Fisher Scientific Inc., Waltham, MA U.S.A.) and separated using a 130 min gradient (100 min from 0% -35% Buffer B; 20 min from 35% -45% Buffer B; 5 min from 45% -95% Buffer B; 4min 95% Buffer B and 1 min 0% Buffer B; (Buffer A: 0.1% formic acid, 2% acetonitrile and Buffer B: 0.1% formic acid in acetonitrile) at a flow-rate of 250 nL/min on a nanoEasy HPLC (Proxeon) coupled to a nanoelectrospray (Thermo Fisher Scientific). Mass spectra were acquired on a LTQ-Orbitrap Velos mass spectrometer (Thermo Fisher Scientific) in the positive ion mode. Full-scan MS spectra (m/z 300–18000) were acquired in the Orbitrap at a resolution of 60,000 and the 15 most intense ions were selected for collision induced dissociation (CID) fragmentation in the linear ion trap with a normalized collision energy of 35%. Singly charged ions and unassigned charge states were rejected. Dynamic exclusion was enabled with exclusion duration of 45 s.

MS data were analysed according to [19], Mass spectra raw files were searched against an in–house specific database with known venom allergen sequences and sequences extracted from transcriptomic databases, using the Sequest search engine through Proteome Discoverer (version 1.4.1.14) (Thermo Fisher Scientific). Search parameters included a maximum of two missed cleavages allowed, carbamidomethyl of cysteines as a fixed modification and oxidation of methionine as variable modifications. Precursor and fragment mass tolerance were set to 10 ppm and 0.5 Da, respectively. Identified peptides were validated using Percolator algorithm with a q-value threshold of 0.01. The data generated by these procedures, that has allowed the characterization of the *Vespa velutina* components included in this manuscript are available at the Mass spectrometry Interactive Virtual Enviroment (MassIVE), with the reference PXD015381 as ProteomeExchange identifier.

The publicly available transcriptomic data for the Asian hornet *Vespa velutina nigrithorax* [7] was downloaded from the GEO database (SRA experiment SRX595647). The sequence data was originally obtained from RNA extracted from the venom glands of Asian hornet species collected in China ([7]). Using the FASTX-Toolkit (http://hannonlab.cshl.edu/fastx_toolkit/), the sequencing reads were trimmed and filtered with respect to quality, trimming bases from the ends with quality score below 20 and keeping reads that were at least 50 bases long after trimming, resulting in 203,886,099 reads (99.9% of the original data set). *De novo* transcriptome assembly was performed using the Trinity software (v2.1.1) [20]. From the assembled transcript sequences, the "coding sequence" or CDS (sequences corresponding to the natural proteins in the Open Reading Frames) can be extracted for further analyses.

Nomenclature used for the potential allergenic molecules of *Vespa velutina* venom, follow the rules recommended by the WHO/IUIS Committee ([11,12]). Protein parameters from the protein sequences (e.g. molecular weight calculation, extinction coefficients, prediction of N-glycosylation, etc.) and alignments were obtained from the software package at the Expasy Bioinformatic portal at the SIB (Swiss Institute of Bioinformatics) [21]. Furthermore, sequence homology searches were performed with the program BLAST (Basic Local Alignment Tool) [22]) provided by the NCBI (National Centre of Biotechnology Information, U.S. National Library of Medicine, Bethesda, USA).

## Results and discussion

Vesp v 1, the A1-phospholipase from the venom sac extract was purified as in [13], by directly using as specific affinity chromatography (Fig 1) followed by an ion exchange chromatography (Mono-S column, GE Healthcare, Upsala, Sweden). The flowthrough (FT) from the previous affinity chromatography was applied onto a Heparin-Sepharose column (GE-Heathcare,

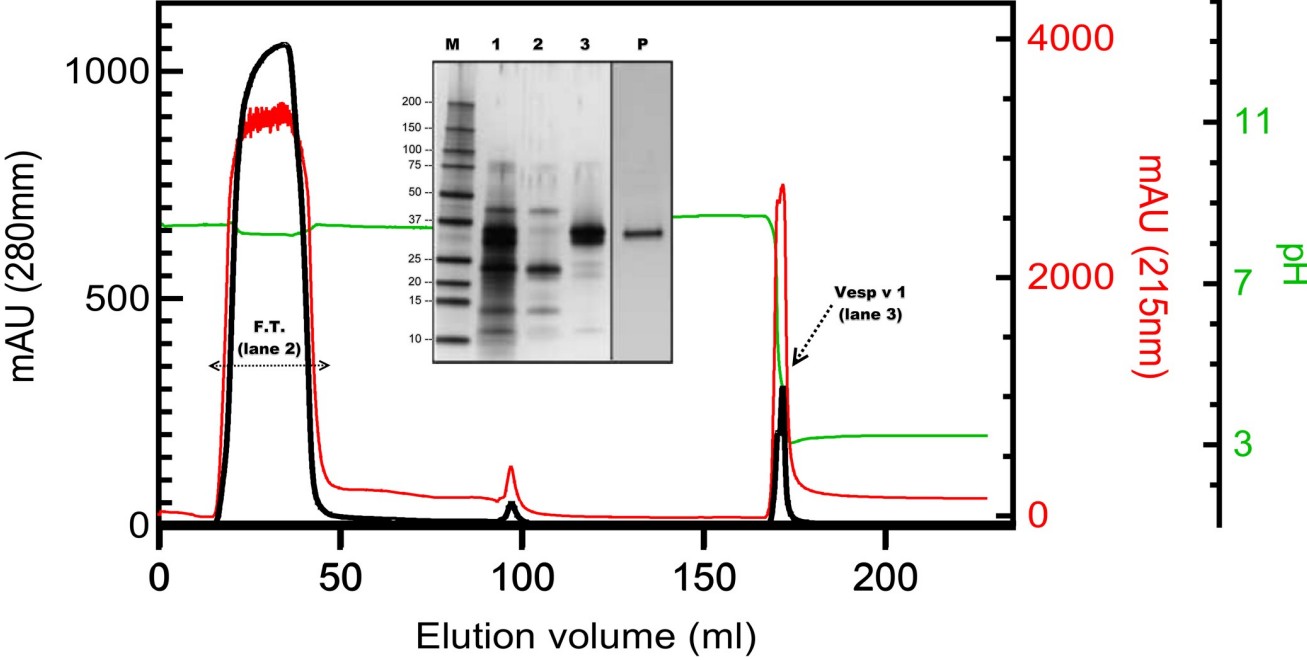

**Fig 1. Elution profile of PC-Sepharose and SDSPAGE analysis.** This affinity chromatography corresponds to the initial separation of the components of the *Vespa velutina* extract. Absorbance at 280nm (black line) and 215nm (red line) show the eluting components ("mAU" corresponds to milli-absorbance units). The second right axis corresponds to pH during the elution (green line), and it can be observed that the phospholipase elution occurs when the pH decreases (the Vesp v 1 peak, as indicated by an arrow). The rest of the venom components elute in the "Flowthrough" (F.T., shown with a horizontal double-arrow). The SDSPAGE in the inset shows the proteins separated in these steps: Lane 1 corresponds to the initial extract, lane 2 corresponds to the FT, and lane 3 to the Vesp v 1 peak. Lane P shows the purified Vesp v 1 (0.25µg), after the final ion-exchange chromatography on a mono-S column. Lane M corresponds to the molecular weight markers (BioRad Precision Blue, BIORAD, Hercules, CA U.S.A.), and their size in kDa are shown on the left margin of the SDSPAGE inset. SDSPAGE shown are silver stained ([15]).

Upsala, Sweden) (Fig 2), which permits the separation of Vesp v 5 and of two isoforms of Hyaluronidase (namely, Vesp v 2A and 2B). Final purification of Vesp v 5 is achieved by an additional Size-Exclusion-Chromatography (SEC) on a Superdex75 column (GE Helthcare, Upsala, Sweden). In the chromatographic profile shown in Fig 2, and additional test was performed in some fractions: the measurement of the relative specific activity of hyaluronidases by the method of Richman ([17]). These measurements just show the presence of hyaluronidase in such fractions, and this can only be considered as a semi-quantitative approach. The data in Fig 2 show the wheal size in the agarose plate and the fractions in which there is a higher (***) or lower (*) hyaluronidase activity (compared with the positive control used in these assays, as indicated in [17]). Moreover, the data shown in the SDSPAGE inserted in Fig 2, clearly shows the presence of the two hyaluronidase isoforms (mainly in fractions c, d and e), as well of the importance of the final SEC step for purifying Vesp v 5.

The proteins purified have been analyzed by direct N-terminal protein sequencing (Edman degradation), showing the results presented in Table 1. In the case of Vesp v 2, it is clear that there are two proteins (confirming what it was mentioned in the SDSPAGE of Fig 2), and their sequences correspond to the first 10 residues of each of the isoforms found in the *Vespa velutina* transcriptome databases (Fig 3); moreover, the experimental data show that one of the isoforms is clearly the most abundant (isoform B), and this is coincident with what has also been described for the corresponding isoform of Ves v 2 (allergen component from *Vespula vulgaris*) [23]. The data obtained in the N-terminal sequencing also demonstrates that the Asn in 3[rd] position of isoform A must be glycosylated (wasps and hornet hyaluronidases are known to be

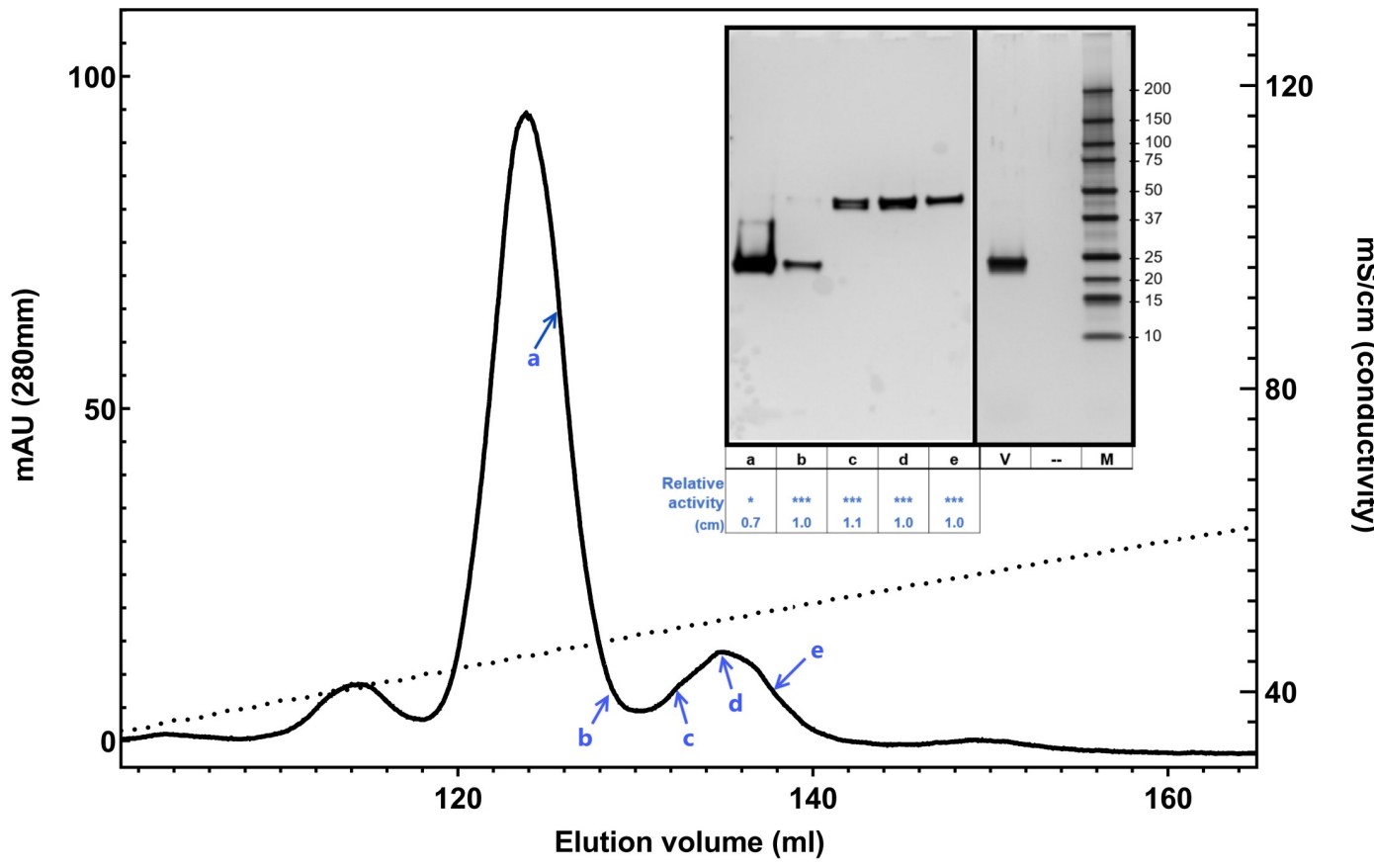

**Fig 2. Elution profile of Heparin-Sepharose column and SDSPAGE analysis.** This chromatographic profile shows the separation of the components coming from the FT of the chromatography in Fig 1 (absorbance at 280nm represented as ________) due to the effect of the NaCl gradient (conductivity, in mS/cm, represented as . . . . . .. .. . . .): Vesp v 5 elutes as the major peak, clearly separated from the Vesp v 2 hyaluronidases isoforms. In the inserted silver-stained SDSPAGE, the left panel shows consecutive fractions of the chromatography (fractions a, b, c, d, e, whose positions are shown by blue arrows); in addition to the electrophoretic result, the inserted table shows the relative hyaluronidase activity measured in those fractions (expressed in cm, as the size of the wheal in the agarose plate, or indicating its higher "***" or lower "*" relative activity). Finally, the right panel shows the purified Vesp v 5 (lane V, 0.25μg), a blank lane (--) and molecular weight markers (lane M). The molecular weight in kDa for the BioRad Precision Blue are shown on the right margin of the SDSPAGE inset.

glycosylated; e.g. Ves v 2 in known to contain at least 3 glycosylation sites [24]). This is also confirmed by the apparent molecular weight exhibited by the two isoforms (40.0–41.7kDa, as calculated from the SDS-PAGE, Fig 2), which are higher than the 39.1 and 40.0kDa values that

**Table 1. N-terminal sequences determined for the purified proteins of the venom of *V.velutina*.** The sequential result of the amino acids resulting from the Edman degradation are shown (in the case of Vesp v 1 and Vesp v 5 only 5 sequencing cycles were performed). In the column at right, the corresponding accession numbers assigned to each protein in the UniProt Knowledgebase are shown. The purified proteins are named considering their potential allergenic nomenclature, even if the only major allergen confirmed to date is Vesp v 5, as approved by the WHO/IUIS Committee ([11,12]), and included in their allergen database at "www.allergen.org".

| Protein | N-terminal sequencing results | | | | | | | | | | UNIPROT # |
|---|---|---|---|---|---|---|---|---|---|---|---|
| Vesp v 2A | Asn | Leu | (*) Asn | Arg | Thr | Asn | Trp | Pro | Lys | Lys... | C0HLL4 |
| Vesp v 2B | Ser | Glu | Arg | Pro | Lys | Arg | Val | Phe | Asn | Ile... | C0HLL5 |
| Vesp v 1 | Gly | Leu | Leu | Pro | Lys... | | | | | | C0HLL3 |
| Vesp v 5 | Asn | Asn | Tyr | Cys | Lys... | | | | | | P0DMB9 |

(*) Asn derivative corresponding to N-glycosylation

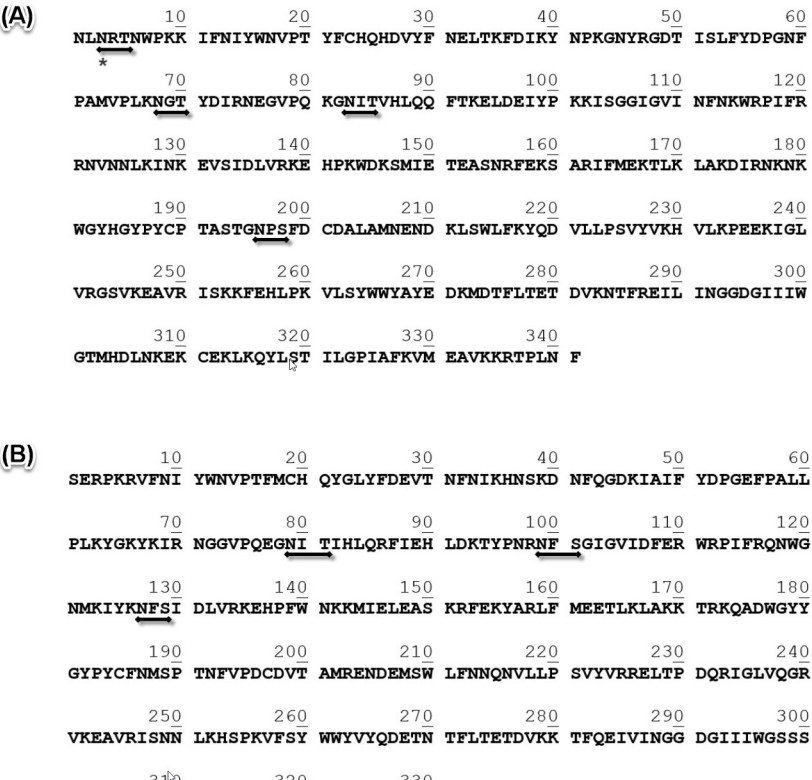

**(A)**

```
        10         20         30         40         50         60
NLNRTNWPKK IFNIYWNVPT YFCHQHDVYF NELTKFDIKY NPKGNYRGDT ISLFYDPGNF
*
        70         80         90        100        110        120
PAMVPLKNGT YDIRNEGVPQ KGNITVHLQQ FTKELDEIYP KKISGGIGVI NFNKWRPIFR

       130        140        150        160        170        180
RNVNNLKINK EVSIDLVRKE HPKWDKSMIE TEASNRFEKS ARIFMEKTLK LAKDIRNKNK

       190        200        210        220        230        240
WGYHGYPYCP TASTGNPSFD CDALAMNEND KLSWLFKYQD VLLPSVYVKH VLKPEEKIGL

       250        260        270        280        290        300
VRGSVKEAVR ISKKFEHLPK VLSYWWYAYE DKMDTFLTET DVKNTFREIL INGGDGIIIW

       310        320        330        340
GTMHDLNKEK CEKLKQYLST ILGPIAFKVM EAVKKRTPLN F
```

**(B)**

```
        10         20         30         40         50         60
SERPKRVFNI YWNVPTFMCH QYGLYFDEVT NFNIKHNSKD NFQGDKIAIF YDPGEFPALL

        70         80         90        100        110        120
PLKYGKYKIR NGGVPQEGNI TIHLQRFIEH LDKTYPNRNF SGIGVIDFER WRPIFRQNWG

       130        140        150        160        170        180
NMKIYKNFSI DLVRKEHPFW NKKMIELEAS KRFEKYARLF MEETLKLAKK TRKQADWGYY

       190        200        210        220        230        240
GYPYCFNMSP TNFVPDCDVT AMRENDEMSW LFNNQNVLLP SVYVRRELTP DQRIGLVQGR

       250        260        270        280        290        300
VKEAVRISNN LKHSPKVFSY WWYVYQDETN TFLTETDVKK TFQEIVINGG DGIIIWGSSS

       310        320        330
DVNSLSKCMR LREYLLTVLG PIAVNVTEAV N
```

**Fig 3. Complete coding sequence of hyaluronidases isoforms (Vesp v 2A and Vesp v 2B).** Panels A and B show the complete CDS of extracted from the transcriptomic data of *Vespa velutina* ([7,20]) for Vesp v 2A (341 amino acids, 40012.2 Da) and Vesp v 2B (331 amino acids, 39109.7 Da), respectively. Numbers above the sequences facilitate the localization of the amino acids. In both panels, horizontal lines with arrows located below the sequences indicate the "potential N-glycosylation sites" for each sequence (defined as Asn-Xaa-Ser/Thr sequons, according to the prediction performed at the NetNGlyc 1.0 Server [21]). The only confirmed glycosylated residue (marked with an asterisk below the sequence) is Asn-3 in the Vesp v 2A isoform, as shown in the data presented in Table 1. Besides, in the referred table, the accession numbers assigned to these sequences in the UNIPROT database are shown.

correspond to the polypeptide part of the complete "coding sequence" or CDS (Fig 3) of isoforms A and B, respectively (Protein Parameters determined in [21]). Fig 3 also shows the positions of the predicted glycosylation sites for each isoform of Vesp v 2 (four potential sites in each of them).

The N-terminal sequence data for both Vesp v 1 and Vesp v 5 show a unique sequence, confirming its purity. Besides their direct analysis by nLC-MS/MS also confirm that most of the peptides found in these analyses correspond exactly to the sequences extracted from the *Vespa velutina* transcriptomic data ([7]), as shown in Fig 4. As it can be observed, 72% and 87% coverage, for Vesp v 1 and Vesp v 5 respectively, have been experimentally determined by this combination of techniques. An interesting feature for Vesp v 1, is that it is a glycosylated phospholipase; this can be deducted from the fact that nLC-MS/MS studies have not provided any data on the peptide in which Asn-61 is potentially predicted as N-glycosylated (Fig 4A); on the contrary, Asn-40 is not glycosylated, since the peptide containing this amino acid is clearly identified by nLC-MS/MS (Fig 4A). This is therefore another peculiar A1-phospholipase, since they are not usually glycosylated (so far, this had been only described for two phospholipases: Dol m I from *Dolichovespula maculata* ([25]) and for Sol i 1, from the imported fire ants ([26]). Moreover, the glycosylation of Vesp v 1 is also confirmed by the higher apparent

**(A)**

```
          10         20         30         40         50         60
GLLPKCKLVP EQISFILSTR ENRNGVFLTL DSLKKGGILN KSDLSSTQVV FLIHGFISSA

          70         80         90        100        110        120
NNSNYMDMTK ALLEKNDCMV ISIDWRNGAC TNEFQILKFI GYPKAVENTR TVGKYIADFS
*
         130        140        150        160        170        180
KLLMQKYKVS LANIRLIGHS LGAQIAGFAG KEYQKFKLGK YPEIIGLDPA GPLFKSNDCS

         190        200        210        220        230        240
QRICETDAHY VQIIHTSNNL GTERTLGTVD FYMNNGYNQP GCYYSFIGET CSHTRAVQYF

         250        260        270        280        290        300
TECIRHECCL IGVPQSKNPQ PVSKCTRNEC VCVGLNAKRY PKTGSFYVPV ESKAPYCNNK

GKKI
```

**(B)**

```
          10         20         30         40         50         60
NNYCKIKCRS GIHTLCKYGT STKPNCGRSV VKASGLTKAE KLEILKQHNE FRQKVARGLE

          70         80         90        100        110        120
TRGNPGPQPP AKSMNTLVWN DELAQIAQVW ASQCKYGHDN CRNTAKYLVG QNIAEQSTTA

         130        140        150        160        170        180
ASFEPVSNMV KMWSDEVKDY QYGSSKNKLN DVGHYTQMVW AKTKEIGCGN IKYIENGWHH

         190        200
HYLVCNYGPA GNIGNEPIYE KK
```

**Fig 4. Complete coding sequence of Vesp v 1 and Vesp v 5, and characterization of its sequences.** Panel A shows the Vesp v 1 CDS extracted from the trancriptomic data of *Vespa velutina* (304 amino acids, 33957.2 Da), highlighting in yellow the peptides identified by nLC-MS/MS analysis, and in green the N-terminal sequence determined (see Table 1). Moreover the two potential N-glycosylation sites for Asn ([21]) are also indicated by horizontal arrows, but only the second predicted site seems to be glycosylated (Asn-61, as explained in the text). Panel B shows the CDS for Vesp v 5 (202 amino acids, 22717.7 Da), similarly coloured as in panel B for Vesp v 1; no N-glycosylation is predicted for this antigen 5. These complete sequence data are available at UNIPROT database, with the accession numbers indicated in Table 1. Besides, the data from the MS/MS analyses for each protein are available with the PXD015381 as ProteomeExchange identifier at the MS Interactive Virtual Enviroment (MassIVE).

molecular weight this protein exhibits in an SDSPAGE (36.1kDa, Fig 1), in comparison with the 34.0kDa that correspond to the polypeptide part of its complete CDS (Fig 4).

These complete CDS, corresponding to relevant proteins present in the venom of *Vespa velutina*, have been assigned the corresponding UNIPROT accession numbers, as shown in Table 1, and are now available for further comparison with other homologous venom components described to date. Vesp v 5 exhibits an apparent molecular weight of 23.5kDa in SDSPAGE (Fig 2), close to the 22.7 kDa calculated size from the complete CDS shown in Fig 4 (as expected, antigen 5 is not a glycosylated component).

The major allergen Vesp v 5, and the potential major allergen Vesp v 1, have been purified to homogeneity, and most of their CDS determined (with an 87% and 74% of total coverage, respectively; Fig 4), matching perfectly with the extracted data from the *Vespa velutina* transcriptome [7]. This also confirms that these sequence data, obtained from the RNA extracted from the venom glands of the hornet species present in China [7], match exactly with the experimental data determined from the isolated natural components from the venom from species present in Europe, where the source material was collected. These data are very relevant, since they show that there have not been evolutive variations in the amino acid sequences of these venom components, in spite of the potential regional variations of the Asian hornet in its original location and the analogous components produced in the venom of *Vespa velutina*

species that entered in Europe in the first decade of the year 2000, and have then expanded its presence throughout several European countries. In addition, it is important to mention that the actual sequences that are presented in this manuscript are the only existing isoforms found in the whole *Vespa velutina* transcriptome, i.e. no additional isoforms were found for Vesp v 1, Vesp v 5 or the two Vesp v 2 isoforms.

In the case of the two only hyaluronidases isoforms found (named Vesp v 2A and Vesp v 2B), the complete CDS have not been extensively characterized experimentally, but the exact match of the first 10 amino acid residues of each isoform (Fig 3 / Table 1) permits to reasonably think that the whole CDS will be those extracted from the complete transcriptomic database. Nevertheless, the partial characterization of these hyaluronidases isoforms, is completed, apart from the purity shown in the chromatographic peak containing the aforementioned isoforms (Fig 2), also considering that the enzymatic activity for hyaluronidase was specifically measured in fractions "c", "d" and "e" shown in Fig 2, and as it was the case for the *Vespula vulgaris* hyaluronidase (Ves v 2) purified by Monsalve et al. [10]. Moreover, our data on the Vesp v 2B isoform, as the most abundant hyaluronidase, is coincident with the data presented by Kolarich et al. [23] for the isoforms of *Vespula vulgaris*.

These complete CDS allow an overall sequence comparison with the previously known homologous allergen components of diverse Hymenoptera species, being of interest the similarity with other *Vespa*, *Vespula* or *Polistes* species. A remarkable similarity has been found between the components of *Vespa* and *Vespula*, much higher than with regard to *Polistes*. It is particularly striking, as shown in Fig 5, the fact that one of the hyaluronidase isoforms of *Vespa* and *Vespula* exhibit 93% sequence identity (and only 73% in the case of the isoform present in *Polistes*), and that the other isoform exists in *Vespula* and *Vespa* and it is not present at all in *Polistes* (these data can be confirmed by having compared with the sequences available to the date in the existing public sequence databases of both nucleotide and protein sequences, searched with the program BLAST (Basic Local Alignment Tool) [22]).

The similarities between the different genera, comparing their phospholipases and antigen 5s are shown in Fig 6 and Table 2, and show the highest similarity of the Asian hornet components with those of the *Vespa* and *Vespula* species, in comparison with those of the *Polistes* species.

The availability of the purified components from *V.velutina* venom will allow to confirm their individual relevance as allergens in patients sensitized to *V.velutina* venom. Vesp v 5 had already been described as the major allergen of *V.velutina* venom [6], but the phospholipase Vesp v 1 cannot be discarded also as a major allergen in Hymenoptera-sensitized patients, as previously shown in several studies in which these two allergenic components have been shown to differently affect different allergic patients [10] and also considering the described intrinsic importance as allergens of phospholipases [28]. These pure components (Vesp v 1 and Vesp v 5) have been specifically prepared as "detection reagents" for measuring sIgE, following the procedure described by Erwin et al. [29] and will be tested in patients that have shown anaphylactic shocks after being stung by *V.velutina*.

The clinical data that will be obtained in future experiments with the use of these isolated components, as well as the overall data of the clinical history of patients being already treated with different commercially available preparations, will confirm if the clear similarity existing among *Vespa* and *Vespula* components justifies that the treatment of these patients with *Vespula*-containing preparations is really being effective in protecting these patients, as already shown for *Vespa* spp.-sensitized patients [30,31], that had been protected by immunotherapy with *Vespula* commercial preparations, in contrast with the opinion of other authors who consider that a specific treatment would be more adequate [32], even if they mainly refer to safety issues.

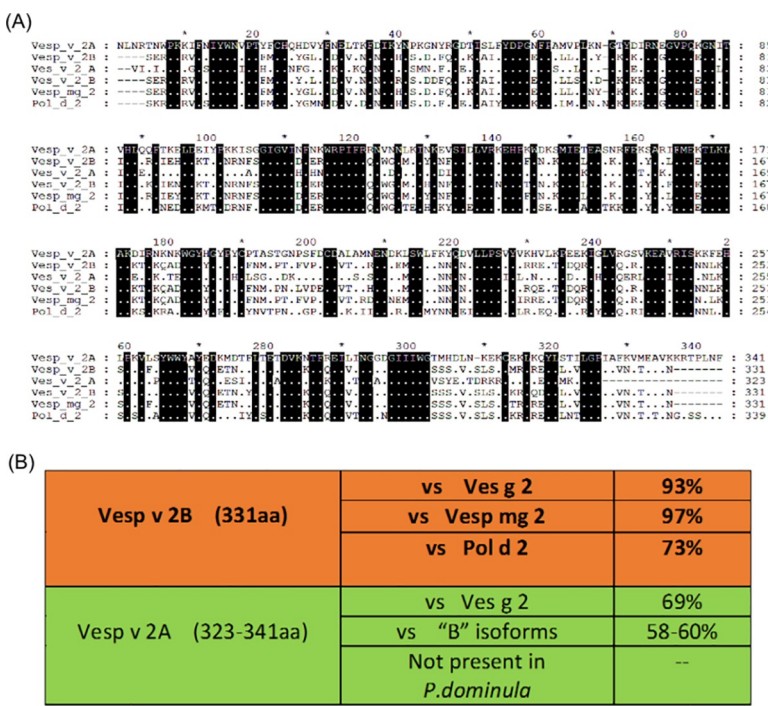

**Fig 5. Alignment of several Hymenoptera hyaluronidases with respect to Vesp v 2 isoforms.** Panel A- Alignment of Vesp v 2 isoforms with the sequences from relevant species available in sequence-databases. The multiple sequence alignment is performed by the program ClustalW ([27]). In the overall alignment, the upper sequence corresponds to Vesp v 2A, and below that, a dot represents identity with the upper sequence, and a dot surrounded in a black square indicates absolute identity among all the aligned proteins (on the other hand, when there is no identity, the corresponding amino acid is shown); finally, the gaps opened by the ClustalW alignment program are represented by "-". On the right margin of the sequences, the corresponding numbering of each sequence is shown. The sequences compared with the two new *Vespa velutina* isoforms, shown as Vesp_v_2A and Vesp_v_2B, and shown in the alignment are: two isoforms from *Vespula germanica*, Ves g 2A (Uniprot Q05FZ1) and Ves_g_2B (Uniprot Q05FZ2), and the hyaluronidases found for *Vespa magnifica* (Vesp_mg_2, Uniprot P86875) and *Polistes dominula* (Pol_d_2; NCBI Reference XP_015179722). In all cases, mature proteins are compared (i.e. starting from the N-terminal of the corresponding protein).Panel B summarizes the identity percentages among these homologous hyaluronidases aligned, differentiating the very high similarities of Vesp v 2B with *Vespa* and *Vespula*'s hyaluronidase (above), and the somehow lower similarities versus (vs) the Vesp v 2A isoform.

More biochemical and immunological studies, including component-resolved-diagnosis (CRD) with the proteins purified in this work and other components available, are therefore needed to confirm which will be the best treatment for these sensitized patients, also considering that the patients being treated nowadays are very probably patients that have become allergic after stings of the wasps and hornets present in these areas until now (*Vespula*, *Polistes* and/or *Vespa crabro*), but there might be a completely new population of patients who will develop allergenicity by *Vespa velutina* stings as the culprit insect.

## Conclusions

Four important components from the venom of *Vespa velutina* (Asian hornet) have been purified and partially characterized. Apart from Vesp v 5, the antigen 5 already described as a major allergen, also the A1-phospholipase and two hyaluronidases isoforms have been purified (they are potentially allergenic components, named Vesp v 1, and Vesp v 2A and Vesp v 2B, respectively).

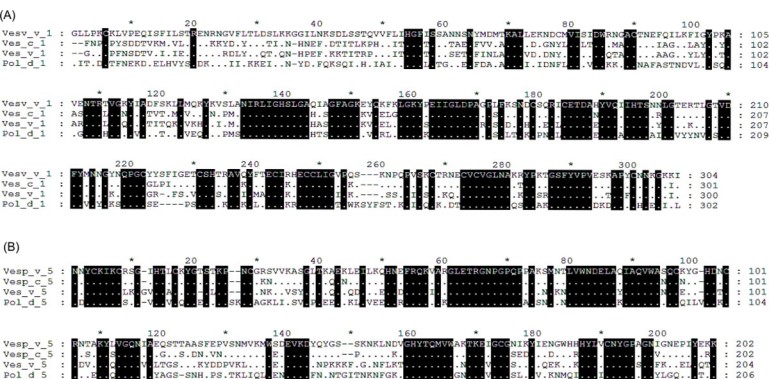

**Fig 6. Multiple sequence alignment of Vesp v 1 and Vesp v 5 with other homologous Hymenoptera components.** Panel (A) and (B) show the alignment of the complete CDS for phospholipases and antigen 5s respectively. The complete mature Vesp v 1 and Vesp v 5 sequences are shown, and are compared with the corresponding mature sequences of a homologous representative from *Vespa crabro*, *Vespula* vulgaris and *Polistes dominula*. The colouring scheme for the alignments is the same than in Fig 5A. On the right margin of the sequences, the corresponding numbering of each sequence is shown. In panel (A) the accession codes of the proteins compared with Vesp v 1 are: Vesp_c_1 (Uniprot P0CH87), Ves_v_1 (Uniprot P49369) and Pol_d_1 (Uniprot Q6Q252). The accession codes of the proteins compared with Vesp v 5 (panel B) are: Vesp_c_5 (Uniprot P35782), Ves_v_5 (EMBL CAB42887.1) and Pol_d_5 (Uniprot P81656).

**Table 2. Sequence identity and similarity of Vesp v 1 and Vesp v 5 in comparison with the homologous components of other Hymenoptera.** Data are obtained after the sequence alignments shown in Fig 6, the values shown correspond to the percent identity, and the percent similarity is shown in brackets (similarity refers to residues having total identity between the sequences, plus those amino acids that are considered to be conservative substitutions as explained in [27]).

| Identiy % (similarity %) with respect to *Vespa velutina* components | | |
|---|---|---|
| | **Phospholipases** | **Antigen 5s** |
| *Vespa crabro* | 69 (78) | 88 (93) |
| *Vespula vulgaris* | 61 (76) | 66 (77) |
| *Polistes dominula* | 53 (71) | 58 (75) |

The complete coding sequences (CDS) of these components are now available, after this characterization and having been extracted their CDS from the transcriptome data available for the venom glands of *V.velutina*.

The purified components will be used to further characterize the sensitization of patients that show anaphylactic shocks after being stung by these Asian hornets, which have invaded several European countries, and treat these patients in the best way possible.

## Supporting information

**S1 Fig. Raw images of SDSPAGEs presented in Figs 1 and 2.** (PDF)

## Acknowledgments

We acknowledge the accurate work performed by the CIB Protein Chemistry Service (CSIC Madrid–Spain), specifically that of Dr.Javier Varela and his team, for the analyses carried out to characterize the purified proteins shown in this work.

## Author Contributions

**Data curation:** Rafael I. Monsalve, Ilka Hoof.

**Investigation:** Rafael I. Monsalve, Ruth Gutiérrez.

**Project administration:** Manuel Lombardero.

**Supervision:** Manuel Lombardero.

**Writing – original draft:** Rafael I. Monsalve.

**Writing – review & editing:** Rafael I. Monsalve, Manuel Lombardero.

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
