## [Decision Letter · Decision Letter 0]

10 Sep 2019

PONE-D-19-22139

Purification and molecular characterization of phospholipase, antigen 5 and hyaluronidases from the venom of the Asian hornet (Vespa velutina)

PLOS ONE

Dear DR. MONSALVE,

Thank you for submitting your manuscript to PLOS ONE. After careful consideration, we feel that it has merit but does not fully meet PLOS ONE’s publication criteria as it currently stands. Therefore, we invite you to submit a revised version of the manuscript that addresses the points raised during the review process.

We would appreciate receiving your revised manuscript by Oct 25 2019 11:59PM. To enhance the reproducibility of your results, we recommend that if applicable you deposit your laboratory protocols in protocols.io, where a protocol can be assigned its own identifier (DOI) such that it can be cited independently in the future. For instructions see: http://journals.plos.org/plosone/s/submission-guidelines#loc-laboratory-protocols

We look forward to receiving your revised manuscript.

Kind regards,

Paulo Lee Ho, Ph.D.

Academic Editor

PLOS ONE

Journal Requirements:

4. Thank you for stating the following in the Financial Disclosure section:

"The author(s) received no specific funding for this work, apart from their regular salary."

We note that one or more of the authors are employed by a commercial company: ALK-Abelló

Reviewers' comments:

Reviewer's Responses to Questions

**Comments to the Author**

1. Is the manuscript technically sound, and do the data support the conclusions?

Reviewer #1: Yes

Reviewer #2: Yes

2. Has the statistical analysis been performed appropriately and rigorously? 

Reviewer #1: N/A

Reviewer #2: N/A

3. Have the authors made all data underlying the findings in their manuscript fully available?

Reviewer #1: Yes

Reviewer #2: Yes

4. Is the manuscript presented in an intelligible fashion and written in standard English?

Reviewer #1: Yes

Reviewer #2: Yes

5. Review Comments to the Author

Reviewer #1: The manuscript is technically sound and needs only minor revisions, which are pointed in pdf file together with a few comments and/or observations.

It is written in understandable English. Highlights should be pointed out by the authors in the Abstract emphasizing the contribution of this study.

All the experiments were carried out with scientific and ethical criteria. The results presented in tables and figures, are essentially important so that the reader can follow all the experimental approaches used, as well as the conclusions that they allowed to arrive. All of them are of quality, demonstrate the results obtained and supported the conclusion established by the authors. Images of electrophoretic gels and blots are free from apparent manipulation.

The results obtained here, open new perspectives to further biochemical and immunological studies with the proteins purified in this work and other components available to confirm which will be the best treatment for sensitized patients, that have become allergic after stings of the wasps and hornets, until a new population of patients who will develop allergenicity by Vespa velutina stings, as the culprit insect.

Reviewer #2: This manuscript just describes the molecular characterization and purification of a phospholipase (Vesp v 1), an antigen 5 (Vesp v 5) and two hyaluronidase isoforms (Vesp v 2A and Vesp v 2B) from the Vespa velutina nigrithorax wasp venom (Hymenoptera, Vespidae).

This wasp is known as yellow-legged Asian Hornet. It Is a species of hornet indigenous to Southeastern Asia. It is of concern as an invasive species in some other countries and in Portugal, it predates honey-bees and other insects.

It has been studied for causing major human accidents in the North-Western parts of the Spanish coast.

In case this paper is approved for publication by PlosOne, I would rather suggest the authors to answer the following questions:

- What is effectively the extent of this wasp on human accidents?

- What is the significance of the translational medicine standpoint for this type of research, I´d say, to mitigate allergic reactions (treatment) and better diagnosis?

- How can differences in sensitization to immune reaction between Vespula and Polistes be explained since they are very close phylogenetically (daughter group), since both belong to the Vespidae family?

- The most important concern of this manuscript is: authors used a pool of wasp venoms from all over Europe. Doesn´t this make it difficult because of individual variations in wasp venoms from different regions of Europe? I mean, antigen detection by antibodies should be specific as well as treatment. We must know exactly which immunogenic compounds are present in a venom gland of a specific species for the detection and later medication. Therefore, if you use a pool, how effective can these two (diagnosis and treatment) factors be, in addition to those molecules evidenced in this study? Is there any study showing that there are differences between venoms of this wasp, due to food, from different regions?

- Is there any specificity and / or similarity between the isolated molecules and those already well-known from other sources of hymenoptera and / or other animals?

- Actually, the discussion of the manuscript is contained from page 335 to 362. Therefore, it is quite powerless and does not, in my view, match the level of the journal.

- Finally, there is no consistent discussion of isolated molecules and effective actions to solve medical and immunological problems.

- Which chromatography method was used: FPLC or HPLC?

- What is a "direct analysis from the protein in solution"?

- No reagents used in the study have the company's city and country of origin.

- The figures are not of good quality and need to be improved.

6. PLOS authors have the option to publish the peer review history of their article (what does this mean?). If published, this will include your full peer review and any attached files.

Reviewer #1: No

Reviewer #2: No

---

## [Author Response · Author response to Decision Letter 0]

11 Oct 2019

RESPONSES HAVE BEEN INCLUDED IN THE DOCUMENT "RESPONSE TO REVIEWERS"

---

## [Decision Letter · Decision Letter 1]

11 Nov 2019

Purification and molecular characterization of phospholipase, antigen 5 and hyaluronidases from the venom of the Asian hornet (Vespa velutina)

PONE-D-19-22139R1

Dear Dr. MONSALVE,

We are pleased to inform you that your manuscript has been judged scientifically suitable for publication and will be formally accepted for publication once it complies with all outstanding technical requirements.

With kind regards,

Paulo Lee Ho, Ph.D.

Academic Editor

PLOS ONE

Additional Editor Comments (optional):

Reviewers' comments:

Reviewer's Responses to Questions

**Comments to the Author**

1. If the authors have adequately addressed your comments raised in a previous round of review and you feel that this manuscript is now acceptable for publication, you may indicate that here to bypass the “Comments to the Author” section, enter your conflict of interest statement in the “Confidential to Editor” section, and submit your "Accept" recommendation.

Reviewer #2: (No Response)

Reviewer #3: All comments have been addressed

2. Is the manuscript technically sound, and do the data support the conclusions?

Reviewer #2: (No Response)

Reviewer #3: Yes

3. Has the statistical analysis been performed appropriately and rigorously? 

Reviewer #2: Yes

Reviewer #3: Yes

4. Have the authors made all data underlying the findings in their manuscript fully available?

Reviewer #2: Yes

Reviewer #3: Yes

5. Is the manuscript presented in an intelligible fashion and written in standard English?

Reviewer #2: No

Reviewer #3: Yes

6. Review Comments to the Author

Reviewer #2: I believe all the corrections made by the authors are quite satisfactory and the present study, now deserves to be published in PlosOne.

Reviewer #3: This manuscript by Rafael et al. aims to purify and identify the potential allergenic components from Vespa velutina venom. The phospholipase, antigen 5 and two hyaluronidase isoforms have been well confirmed and partially characterized in this work. The revised manuscript has profoundly addressed the comments contributed by reviewers. I have no additional comments, and this manuscript is now suitable for publication.

7. PLOS authors have the option to publish the peer review history of their article (what does this mean?). If published, this will include your full peer review and any attached files.

Reviewer #2: No

Reviewer #3: No

---

## [Editor Report · Acceptance letter]

27 Dec 2019

PONE-D-19-22139R1 

Purification and molecular characterization of phospholipase, antigen 5 and hyaluronidases from the venom of the Asian hornet (*Vespa velutina*) 

Dear Dr. MONSALVE:

I am pleased to inform you that your manuscript has been deemed suitable for publication in PLOS ONE. Congratulations! Your manuscript is now with our production department. 

With kind regards,

on behalf of

Dr. Paulo Lee Ho 

Academic Editor

PLOS ONE